# Time series modeling of cell cycle exit identifies Brd4 dependent regulation of cerebellar neurogenesis

Clara Penas[1,8,9], Marie E. Maloof [1,9], Vasileios Stathias[1], Jun Long[2], Sze Kiat Tan [1], Jose Mier[3], Yin Fang[4], Camilo Valdes[5], Jezabel Rodriguez-Blanco[2], Cheng-Ming Chiang[6], David J. Robbins[2], Daniel J. Liebl[3], Jae K. Lee[3], Mary E. Hatten[4], Jennifer Clarke[7] & Nagi G. Ayad[1]

Cerebellar neuronal progenitors undergo a series of divisions before irreversibly exiting the cell cycle and differentiating into neurons. Dysfunction of this process underlies many neurological diseases including ataxia and the most common pediatric brain tumor, medulloblastoma. To better define the pathways controlling the most abundant neuronal cells in the mammalian cerebellum, cerebellar granule cell progenitors (GCPs), we performed RNA-sequencing of GCPs exiting the cell cycle. Time-series modeling of GCP cell cycle exit identified downregulation of activity of the epigenetic reader protein Brd4. Brd4 binding to the *Gli1* locus is controlled by Casein Kinase 1δ (CK1 δ)-dependent phosphorylation during GCP proliferation, and decreases during GCP cell cycle exit. Importantly, conditional deletion of Brd4 in vivo in the developing cerebellum induces cerebellar morphological deficits and ataxia. These studies define an essential role for Brd4 in cerebellar granule cell neurogenesis and are critical for designing clinical trials utilizing Brd4 inhibitors in neurological indications.

[1] Department of Psychiatry and Behavioral Sciences, Center for Therapeutic Innovation, Sylvester Comprehensive Cancer Center, Miami Project to Cure Paralysis, University of Miami Miller School of Medicine, Miami, FL 33136, USA. [2] Department of Surgery, University of Miami Miller School of Medicine, Miami, FL 33136, USA. [3] Department of Neurosurgery, Miami Project to Cure Paralysis, University of Miami Miller School of Medicine, Miami, FL 33136, USA. [4] Laboratory of Development Neurobiology, The Rockefeller University, New York, NY 10065, USA. [5] Computing and Information Sciences, Florida International University, Miami, FL 33199, USA. [6] Simmons Comprehensive Cancer Center, Department of Biochemistry and Department of Pharmacology, University of Texas Southwestern Medical Center, Dallas, TX 75390, USA. [7] Department of Statistics, University of Nebraska, Lincoln, NE 68588, USA. [8] Present address: Institut de Neurociències, Departament de Biologia Cellular, Fisiologia i Immunologia, Universitat Autònoma de Barcelona, CIBERNED, Bellaterra 08193, Spain. [9] These authors contributed equally: Clara Penas, Marie E. Maloof. Correspondence and requests for materials should be addressed to N.G.A. (email: nayad@miami.edu)

During postnatal mammalian development, granule cell progenitors (GCPs) undergo symmetric divisions in the external germinal layer (EGL) of the brain and exit the cell cycle within a narrow time frame, resulting in rapid cellular expansion and differentiation[1]. However, the fundamental mechanisms controlling irreversible GCP cell-cycle exit have not been elucidated. It is essential we discover these mechanisms to understand cerebellar development, as defects in GCP expansion have been linked to cerebellar ataxia and the most common pediatric brain tumor, medulloblastoma[2]. Epigenetic modifiers control gene expression without changing DNA sequence. These modifiers include histone acetyltransferases (HATs) and histone deacetylases (HDACs). Histone acetyltransferases attach acetyl groups to lysine residues on histone proteins, while HDACs remove those modifications[3]. Histones normally bind DNA molecules via their positively charged lysine and arginine tails[3]. Histone–DNA binding initiates DNA compaction and transcriptional silencing. Histone tail acetylation reduces this positive charge, attenuates DNA binding and compaction and allows transcription. Histone deacetylation has the opposite effect, thus allowing histone–DNA binding and reducing transcription. Working in combination with HATs (writers) and HDACs (erasers) are histone-reader proteins, which bind to acetylated lysines on histones, recruit transcriptional complexes, and mediate gene transcription. Among the "reader" proteins are bromodomain and extra-terminal domain (BET) proteins. BET proteins include Brd2, Brd3, Brd4, and BrdT. Tissue expression studies suggest that Brd4 is heavily expressed in the brain, making it a likely candidate for modulating neurogenesis in the nervous system[4]. Recent sequencing studies have implicated many epigenetic regulators in medulloblastoma[5]. The epigenetic reader protein Brd4 has been implicated in various cancers, including medulloblastoma[6–13]. Brd4 controls expression of the medulloblastoma essential gene *MYC* in G3 medulloblastomas, which have poor prognosis as well as *GLI1* and *GLI2* levels in Sonic hedgehog (SHH)-driven medulloblastomas, which have intermediate prognosis. Highly selective Brd4 inhibitors have been developed that reduce *MYC*, *GLI1*, and *GLI2* levels. These inhibitors have gone into clinical trials for multiple cancer indications, and one Brd4 inhibitor has received fast-track designation from the FDA for myelofibrosis[14,15]. However, it is unclear whether these inhibitors can be given to children suffering from medulloblastoma, as we do not fully understand the role of Brd4 during normal development. To address this issue, we deleted Brd4 in the developing cerebellum in mice and find that it is essential for cerebellar growth. Brd4 knockout leads to cerebellar ataxia that is linked to defects in cerebellar development starting at postnatal day 3. These studies suggest that Brd4 inhibitors may need to be given during a short developmental window in children to reduce potential negative effects on cerebellar development.

## Results

### Brd4 phosphorylation decreases during cell-cycle exit of granule cell progenitors.
GCPs undergo cell-cycle exit and differentiation when plated on poly-D-lysine/laminin coated plates. We utilized an in vitro system to isolate purified GCPs at various times during differentiation and performed RNA sequencing on purified cell populations from postnatal day (P) 6 mice to determine their characteristics during the exit process (Fig. 1). We found that GCPs exited the cell cycle within 24 h of plating as judged by PI-FACS analysis (Fig. 1a) and EdU incorporation (Fig. 1b). The mRNA expression of several proliferation markers decreased during this time period while those of differentiation markers increased (Fig. 1c). For example, the levels of the cell-

cycle regulator, *Cyclin b1*, and the bHLH transcription factor important for maintaining GCPs in a proliferative state, *Atoh1*, decreased by 24 h post plating. By contrast, neuron-specific class III beta-tubulin (*Tuj1*) and the axonal growth marker, *Gap43*, increased at 24 h. To determine the exact timing of changes in cellular pathways during cell-cycle exit, we performed short-time-series modeling[16] with gene ontology clustering analysis of all mRNAs expressed at 0, 2, 4, 6, 12, 24, and 48 h after plating (Fig. 1d). Interestingly, downregulation of cell proliferation pathways (Cluster #47, Fig. 1d; Supplementary Fig. 1, Supplementary Data 1) occurred at the same time as upregulation of neuronal development pathways (Cluster #75, Fig. 1d; Supplementary Fig. 1, Supplementary Data 1), suggesting that the two processes may be temporally and mechanistically linked (Fig. 1d, Supplementary Fig. 1). Importantly, the SHH pathway, which is a major regulator of GCP expansion[17], is downregulated beginning at 2 h of GCP cell-cycle exit, with the SHH effectors *Gli1* and *Gli2* nearing basal levels by 24 h, which is recapitulated in vivo (Supplementary Figs. 1, 2).

Our prior studies demonstrated that the epigenetic reader protein Brd4 regulates *Gli1* levels in mouse embryonic fibroblasts (MEFs) by directly binding the *Gli1* locus[11]. Brd4 is part of a family of bromodomain and extraterminal domain proteins (BETs) that bind to acetylated lysines on histones and recruit transcriptional complexes to induce transcription of various genes involved in cell proliferation, signaling, and inflammation[18]. To test whether Brd4-dependent regulation of *Gli1* changes during GCP cell-cycle exit, we performed Brd4 chromatin immunoprecipitation (ChIP) analysis in GCPs exiting the cell cycle. Brd4 is expressed in the developing cerebellum during P6-9 when GCPs are proliferating and exiting the cell cycle (Supplementary Fig. 3). As observed in Fig. 2a, Brd4 binding to the *Gli1* locus decreased dramatically within the first 2 h of GCP cell-cycle exit. Brd4 activity has been shown to be regulated by phosphorylation[19] and therefore, we tested whether Brd4 phosphorylation decreases during cell-cycle exit. Brd4 phosphorylation decreases with the same kinetics as Brd4 binding to the *Gli1* locus during cell-cycle exit (Fig. 2b, c; Supplementary Fig. 1).

### Casein Kinase 1δ inhibition or deletion reduces Brd4 binding to the Gli1 locus.
We have previously demonstrated that Casein Kinase 1 delta (CK1δ) is required for GCP proliferation, and its protein levels decrease with the same kinetics as Brd4 phosphorylation during cell-cycle exit[20] (Fig. 2b). Therefore, we hypothesized that CK1δ may control Brd4 phosphorylation. Indeed, in vitro phosphorylation assays using purified Brd4 and CK1δ demonstrated that Brd4 is a CK1δ substrate in vitro (Fig. 2d, e). Earlier studies have shown that Brd4 phosphorylation on serines 492/494 relieves Brd4 autoinhibition, thereby allowing it to bind to chromatin[19]. We found that serines 492 and 494 are necessary for CK1δ-mediated phosphorylation of Brd4 as mutating both to alanine abrogated the ability of CK1δ to phosphorylate Brd4 in vitro (Fig. 2d–f). To determine whether CK1δ controls Brd4 activity in GCPs, we incubated proliferating GCPs with the selective CK1δ inhibitor, SR-1277[21], or vehicle in the presence of SHH and measured Brd4 phosphorylation on serines 492/494 at 24 h (Fig. 3a, b). SR-1277 treatment decreased phospho-S492/494 Brd4 levels relative to the total Brd4 (Fig. 3a, b), which correlated with decreased Brd4 binding to the *Gli1* locus as measured by Brd4 ChIP analysis (Fig. 3c). Importantly, CK1δ inhibition reduced Brd4 association with the *Gli1* locus to a similar extent as treatment with the selective Brd4 inhibitor, I-BET151[22], suggesting that CK1δ activity is required for maintaining Brd4 in an active state (Fig. 3c). In agreement with this, conditional deletion of CK1δ in GCPs in vivo reduced Brd4

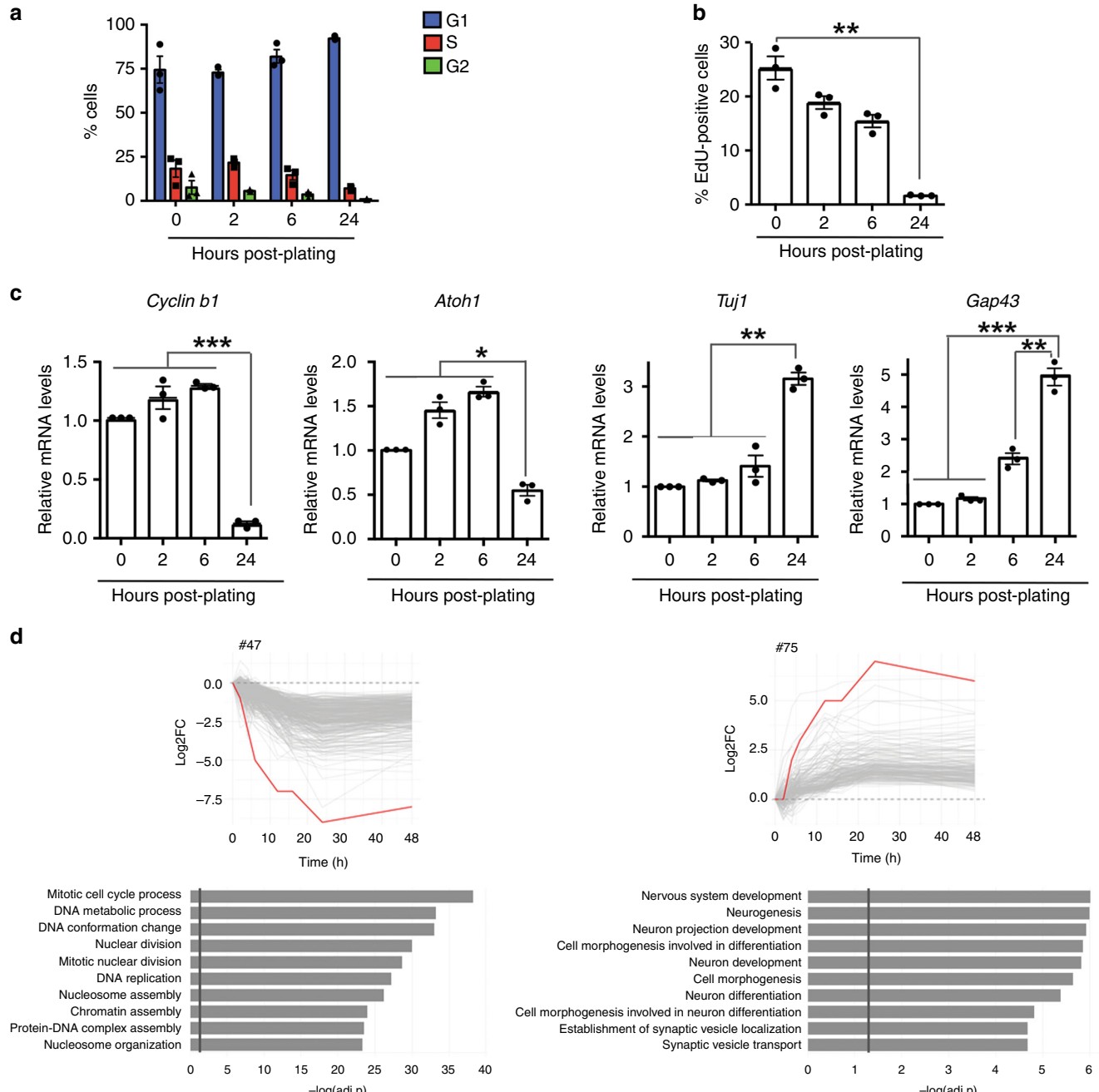

**Fig. 1** Timing of cycle exit in GCPs. **a** GCPs exit the cell cycle within 24 h of plating as judged by PI-FACS analysis. FACS analysis was performed on GCPs purified from P6 mice and plated for the time intervals indicated. FlowJo software was then used to assess the percentage of cells in the G1, S, or G2/M phase. **b** Purified GCPs exit the cell cycle within 24 h of plating as judged by EdU incorporation studies. GCPs purified from P6 mice were processed for EdU incorporation, which was normalized to the total number of cells labeled with Hoechst staining. **c** Purified GCPs from P6 mice were plated and processed for the RNA expression at the time intervals indicated for the proliferative markers, *Ccnb1* and *Atoh1*, and the differentiation markers, *Tuj1* and *Gap43*. qRT-PCR was performed and normalized to *Gapdh*. **d** Short-time series modeling of mRNAs during cell-cycle exit. Plots represent mRNA expression profiles during GCP cell-cycle exit. Representative clusters (#47, #75 and their related cellular processes) are shown. The red line represents the consensus for each cluster. The gray lines represent individual mRNA expression profiles. The associated biological processes as defined by DAVID[41] in each cluster are shown. The identity of each gene in the cluster can be found in Supplementary Data 1. The results are shown as the average values of three independent experiments and are represented as the mean ± SEM. A one-way ANOVA followed by Bonferroni multiple comparison testing was performed (*$p < 0.05$, **$p < 0.01$). Source data can be found in source data graphs under tabs for **a–c**

phosphorylation on serines 492/494 (Fig. 3d), and binding to the *Gli1* locus (Fig. 3e). Consistent with a direct effect on Brd, CK1δ inhibition in Suppressor of Fused[17] deleted (*Sufu*$^{-/-}$) cells that contain active SHH signaling independent of the membrane

receptor Smoothened also reduced *Gli1* levels and Brd4 binding to the *Gli1* locus (Supplementary Fig. 4). Collectively, these studies suggest that CK1δ-mediated phosphorylation of Brd4 on serines 492/494 potentiates Brd4 localization to the *Gli1* locus

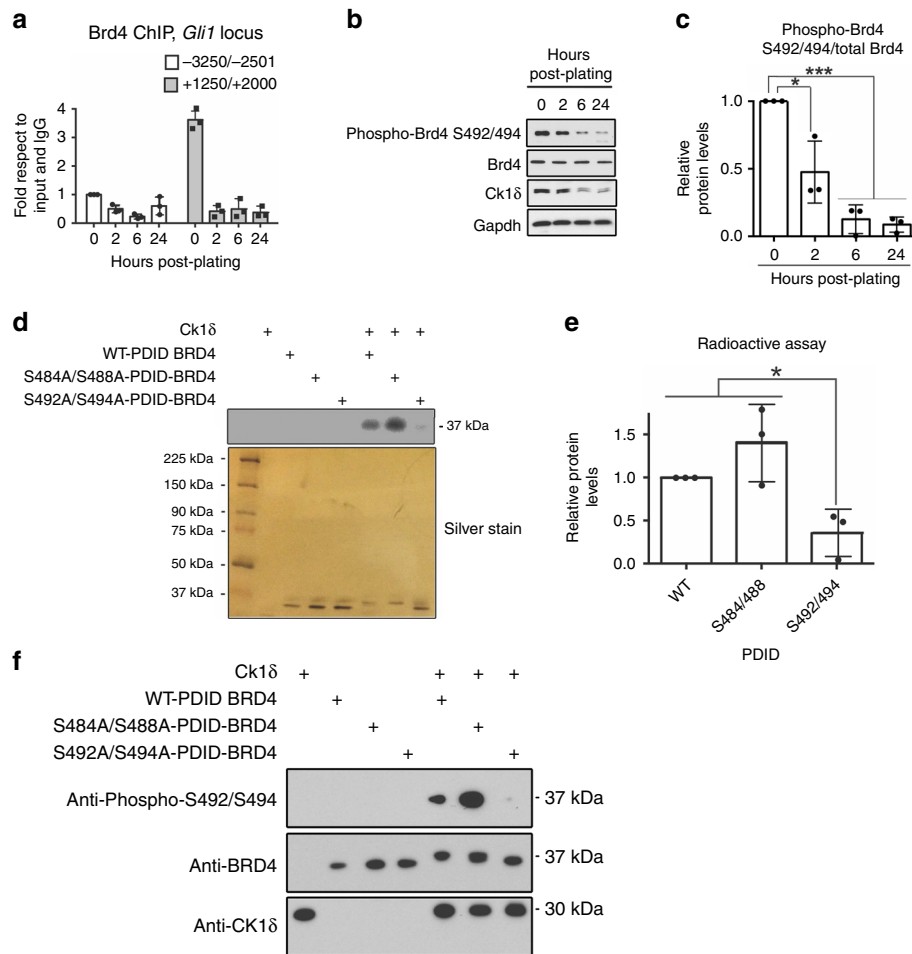

**Fig. 2** Brd4 phosphorylation on S492/494 and binding to the *Gli1* locus is regulated by CK1δ and is reduced during GCP cell-cycle exit. **a** Brd4 association to the *Gli1* locus decreases during GCP cell-cycle exit. White bars represent the association to a region far away from the transcription start site of *Gli1*, and black bars represent the association to a proximal region to the transcription start site (TSS). Brd4 occupancy was analyzed by ChIP-qPCR and then normalized to control ChIP performed using rabbit IgG. **b**, **c** Brd4 phosphorylation at serines 492/494 decreases during GCP cell-cycle exit. **d** CK1δ phosphorylates Brd4 in the PDID domain. The PDID domain of WT-Brd4, S484A/S488A-Brd4, or S492A/S494A-Brd4 were mixed with recombinant CK1δ and $^{32}$P-ATP, and the extent of radioactivity bound to WT or mutant Brd4 was determined after SDS-PAGE and autoradiography. Silver staining showed that recombinant WT-Brd4 and S484A/S488A-Brd4 had reduced electrophoretic mobility in the presence of CK1δ, while the S492 A/S494A-Brd4 mutant migrated slower, indicating lower phosphorylation. Both S492A/S492A are required for efficient phosphorylation of Brd4 by CK1δ. **e** Plot represents the quantification of the phosphorylation levels from the radioactive assay with respect to the total amount of protein quantified from the silver staining. **f** Confirmation of phosphorylation at S492/494 with an anti-phoshpo-Brd4 antibody. Recombinant WT-Brd4, S484A/S488A-Brd4, or S492A/S494A-Brd4 were mixed with recombinant CK1δ and ATP, and the extent of phosphorylation determined by anti-phospho-S492/494 western analysis. Note that CK1δ phosphorylation of WT-Brd4, S484A/S488A-Brd4 leads to signal in the anti-phospho-S492/494 western analysis while no signal is observed with the S492A/S494A-Brd4 mutant, as anticipated. The results are shown as the average values of three independent experiments and are represented as the mean ± SEM. A one-way ANOVA followed by Bonferroni multiple comparison testing was performed (*$p < 0.05$, ***$p < 0.001$). Source data can be found in source data graphs under tabs for **a**, **b**, **e** and in source data Fig. 1 and Fig. 2

during GCP proliferation. By contrast, during GCP cell-cycle exit, rapid downregulation of CK1δ activity is associated with reduced Brd4 phosphorylation on serines 492/494, decreased Brd4 binding to the *Gli1* locus, and attenuated *Gli1* expression.

**Brd4 Inhibition reduces granule cell progenitor proliferation in vitro, ex vivo, and in vivo.** Downregulation of Brd4 phosphorylation and binding during GCP cell-cycle exit suggests that it is an essential regulator of SHH signaling in GCPs, and that disrupting Brd4 activity may limit GCP proliferation. To test this, we measured the effect of BET inhibition on GCP proliferation via the Brd4 inhibitor I-BET151. I-BET151 treatment reduced GCP proliferation and *Gli1* expression in vitro (Fig. 4a, b;

Supplementary Fig. 5). Further, I-BET151 treatment of cerebellar slices ex vivo reduced EdU incorporation in the EGL (Fig. 4c, d). Finally, treatment of P8 pups with a brain penetrant BET inhibitor, JQ1[23], also reduced cerebellar GCP proliferation in vivo (Fig. 4e, f). Collectively, these studies suggest that pharmacological inhibition of Brd4 reduces GCP expansion in the developing cerebellum.

**Brd4 deletion reduces GCP proliferation and induces ataxia.** To determine whether genetic disruption of Brd4 affects cerebellar development, we conditionally deleted Brd4 in GCPs by breeding *Brd4fl/fl* mice to Tg (*Atoh1-Cre*) mice, which express Cre under the Atoh1 promoter after embryonic day 13.5[24] (Fig. 5a). Tg

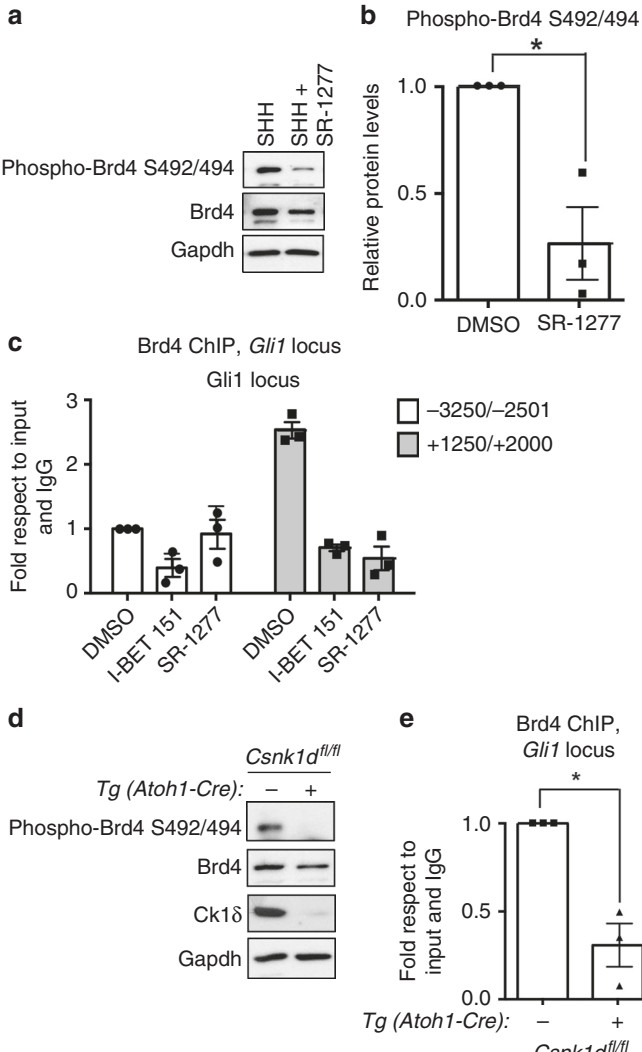

Fig. 3 CK1δ inhibition or deletion reduces Brd4 phosphorylation on S492/494 and binding to the *Gli1* locus. **a** CK1δ is required for Brd4 phosphorylation in vitro. CK1δ inhibition via the highly selective CK1δ inhibitor (SR1277) reduces Brd4 phosphorylation at S492/494. GCPs were isolated from P7 mice and subsequently treated with SHH + DMSO or SHH + SR-1277 for 24 h. Lysates were resolved and subsequently probed for phospho-Brd4 S492/494, total Brd4, or Gapdh control. **b** The ratio of phospho-Brd4 S492/494 to total Brd4 from **a**) was quantified. Note that SR-1277 significantly reduced phosphorylation of Brd4 at S492/494. **c** CK1δ or Brd4 inhibition reduces Brd4 binding to *Gli1* locus, as measured by Brd4 ChIP analysis. GCPs were incubated with either I-BET151 (Brd4 inhibitor) or SR-1277 (CK1δ inhibitor), and the extent of Brd4 bound to the *Gli1* locus was measured after anti-Brd4 ChIP analysis. The amount of Brd4 was normalized to the input DNA and the amount bound using IgG control. DMSO was used as a vehicle control. Note that Brd4 bound close to TSS of *Gli1* and that I-BET151 or SR-1277 was able to displace it from the *Gli1* locus. **d** Conditional deletion of CK1δ in GCPs in vivo reduces phosphorylation of Brd4 at S492/494. CK1δ was deleted in GCPs by breeding *CK1δ^{fl/fl}* mice to *Tg (Atoh1-Cre)* mice and the extent of phospho-Brd4 levels determined via western analysis. The total Brd4 and Gapdh were included as loading controls. Note that CK1δ is efficiently deleted in *Tg (Atoh1-Cre +);CK1δ^{fl/fl}* mice and that phosphorylation at serines S494/494 was lower in these animals relative to *Tg (Atoh1-Cre-);CK1δ^{fl/fl}* mice. **e** CK1δ knockout reduces Brd4 binding to the *Gli1* locus. GCPs were purified from *Tg (Atoh1-Cre +); CK1δ^{fl/fl}* mice or *Tg (Atoh1-Cre-);CK1δ^{fl/fl}* mice, and the amount of Brd4 bound to the *Gli1* locus was measured. Note that CK1δ deletion reduced Brd4 binding to the *Gli1* locus. The results are shown as the average values of three independent experiments and are represented as the mean ± SEM. A paired *t* test was performed (*$p < 0.05$). Source data can be found in source data graphs under tabs for **b**, **c**, **e** and in source data Fig. 3 and Fig. 4

(*Atoh1-Cre +);Brd4^{fl/fl}* or *Tg (Atoh1-Cre-);Brd4^{fl/fl}* mice were attained, and Brd4 expression in isolated GCPs was analyzed via qRT-PCR and western blot analysis. As seen in Fig. 5b, c, Brd4 mRNA and protein levels were lower in *Tg (Atoh1-Cre +);Brd4^{fl/fl}* mice relative to *Tg (Atoh1-Cre−);Brd4^{fl/fl}* mice, suggesting that efficient deletion of Brd4 occurs upon Cre expression. Reduced Brd4 was correlated with lower levels of the positive effectors of the SHH pathway *Gli1 and Gli2*, but not the negative regulator *Gli3* (Fig. 5b). Furthermore, the level of the cell-cycle protein and Gli1 target, cyclin D1, was decreased in GCPs from *Tg (Atoh1-Cre +);Brd4^{fl/fl}* mice relative to *Tg (Atoh1-Cre−);Brd4^{fl/fl}* mice, suggesting that Brd4 is an essential regulator of cell pro-liferation in GCPs (Fig. 5c; Supplementary Fig. 6). Although other BET proteins are expressed during GCP proliferation (Supplementary Fig. 3), they do not compensate for Brd4 loss (Supplementary Fig. 6). Indeed, GCPs from *Tg (Atoh1-Cre +); Brd4^{fl/fl}* mice proliferated less than those from *Tg (Atoh1-Cre−); Brd4^{fl/fl}* mice (Fig. 5d, Supplementary Fig. 6). Consistent with decreased GCP proliferation, cerebella from *Tg (Atoh1-Cre +); Brd4^{fl/fl}* mice were smaller than those from *Tg (Atoh1-Cre-); Brd4^{fl/fl}* mice (Fig. 6a, b) as well as had abherant cerebellar layer formation that persisted throughout early postnatal development (Fig. 6b; Supplementary Figs. 7, 8). Importantly, reduced cerebellar size correlated with behavioral deficits in *Tg (Atoh1-Cre +);Brd4^{fl/fl}* mice, which exhibited symptoms of cerebellar ataxia not evident

in *Tg (Atoh1-Cre−);Brd4^{fl/fl}* mice (Fig. 6c; Supplementary Movies 1–3). Taken together, these studies demonstrate that Brd4 is an essential regulator of GCP proliferation and cerebellar devel-opment in vivo.

## Discussion

We report a novel in vivo function for the epigenetic-reader protein Brd4. Brd4 controls granule cell progenitor expansion in the developing cerebellum. Brd4 deletion leads to defects in cerebellar morphology, which leads to ataxia. Brd4 activity is temporally regulated during cerebellar granule cell development, as both CK1δ-dependent phosphorylation of Brd4 on serines 492/494 and Brd4 binding to the *Gli1* locus decrease during GCP cell-cycle exit. Multiple transcriptional and posttranslational mechanisms have been shown to be required for cell-cycle exit in GCPs in the developing cerebellum[25–28]. To our knowledge, this is the first report demonstrating that the activity of an epigenetic reader pro-tein is modulated during cell-cycle exit in the developing nervous system. Interestingly, we find that Brd4 loss from the *Gli1* locus occurs early during cell-cycle exit (within 2 h), suggesting that it may be an initiating event in the differentiation of cerebellar granule cells. Future studies are needed to determine how temporally con-trolling Brd4 activity is linked with phosphorylation, epigenetic, and ubiquitin pathways that induce cell-cycle exit in the developing cerebellum[25–28]. Interestingly, Brd4 phosphorylation down-regulation during cell-cycle exit correlates with decreases in *Gli2* levels in a cluster of cell-cycle genes that decreases after postnatal day 7 in vivo (Supplementary Fig. 2)[29].

We have defined an essential role for Brd4 in the developing cerebellum. Although other studies have showed that Brd4 is involved in learning and memory later in development and other developmental or cellular processes[4,23,30–37], ours is the first to

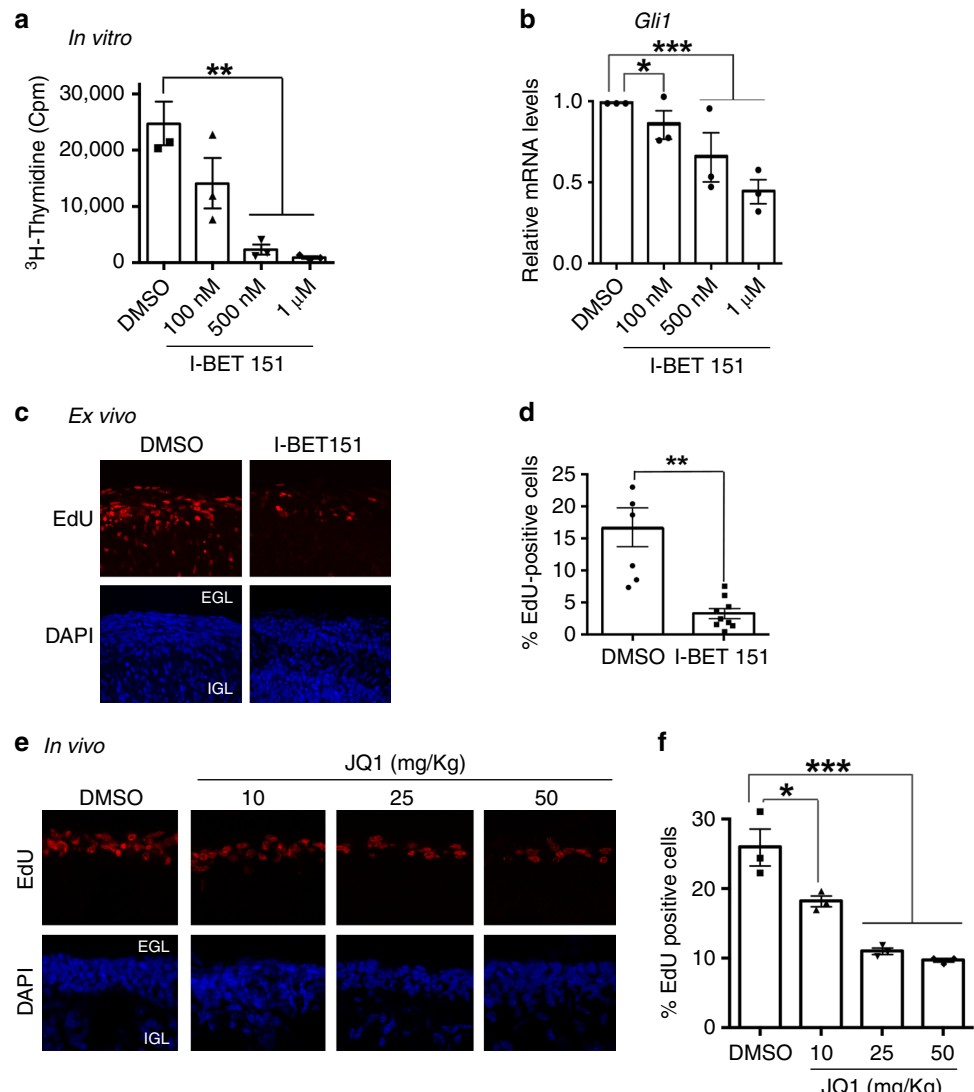

**Fig. 4** BET bromodomain protein inhibition reduces *Gli1* mRNA levels in purified GCPs and GCP proliferation in vitro, ex vivo, and in vivo. **a** BET protein inhibition reduces GCP proliferation in vitro. GCPs were purified from P6 mice and incubated with the indicated concentrations of I-BET151 or DMSO control for 24 h. Proliferation was then measured using a $^3$H-thymidine incorporation assay. **b** BET protein inhibition reduces GCP *Gli1* mRNA levels in vitro. Purified GCPs were incubated as described in **a**. qRT-PCR was performed and normalized to *Gapdh*. **c** Cerebellar organotypic slices were incubated in the presence of either DMSO or 1 μM I-BET151 for 24 h, after which time EdU was added for 2 h. Images were then acquired with a confocal microscope and quantified using ImageJ. **d** Quantification of **c**. **e** Increasing doses of JQ1 yielded a gradual decrease in EdU incorporation in vivo in P8 pups. JQ1 was given to pups for 1 day at 10, 25, or 50 mg/kg every 12 h. Then EdU was injected subcutaneously 1 h prior to perfusion. **f** Quantification of **e**. The results are shown as the average values of three independent experiments and are represented as the mean ± SEM. A one-way ANOVA followed by Bonferroni multiple comparison testing (**a**, **b**, **f**) or a paired *t* test (**d**) was performed (*$p < 0.05$, **$p < 0.01$, ***$p < 0.001$). Source data can be found in source data graphs under tabs for **a**, **b**, **d**, **f**

demonstrate that Brd4 is required for cerebellar growth. Brd4 is required for GCP proliferation by controlling the SHH pathway effectors, *Gli1* and *Gli2*. Brd4 inhibition or deletion reduces GCP proliferation and responsiveness to SHH signaling. Brd4 deletion decreases cerebellum size and induces symptoms of cerebellar ataxia either as a direct effect of neuron loss or indirectly through the aberrant cerebellum morphology (Fig. 6). Brd4 deletion reduces levels of *Gli2*, an essential gene required for vertebrate development[32,33]. Therefore, Brd4's essential role in cerebellar granule cell development is likely to be related to *Gli2* activity required for proper GCP expansion. Brd4, like Gli2, is a target in developmental diseases and cancer, and Brd4 inhibitors may be useful in clinical settings in children[10–13]. However, cerebellum-associated developmental disorders and pediatric cerebellar

tumors such as medulloblastoma are associated with deficits in procedural learning[38–40], which could be exacerbated with prolonged Brd4 inhibitor usage. Our findings suggest that therapeutic use of Brd4 inhibitors may need to be given during a temporal window, as Brd4 may be required for proper cerebellar development in humans.

## Methods

**Animal husbandry**. All mice were housed in an American Association of Laboratory Animal Care–accredited facility at the University of Miami and were maintained in accordance with NIH guidelines. Animal use was approved by the Institutional Animal Care and Use Committee of the University of Miami.

**GCP isolation and compound treatment**. GCPs were purified from cerebellar cortex of P6-8 CD1 and Tg *(Atoh1-Cre +);CK1δ^{fl/fl}* mice by using Percoll gradient

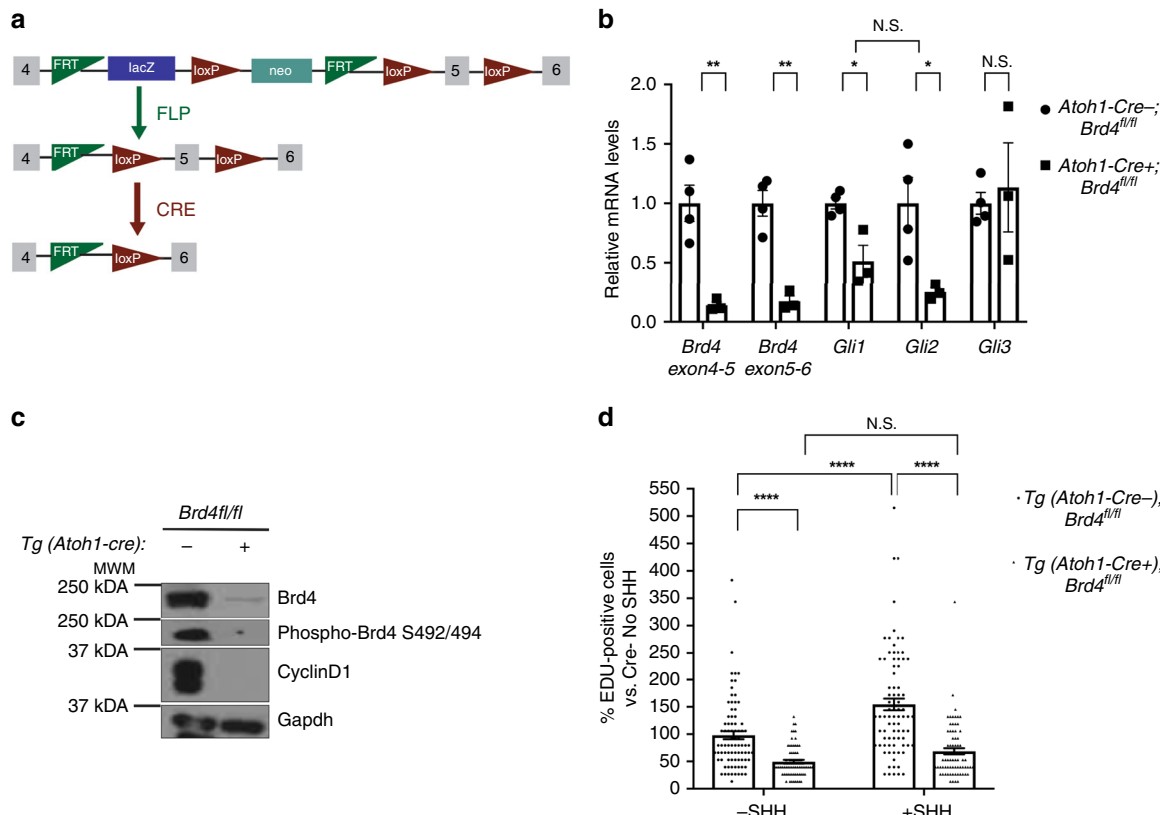

**Fig. 5** Conditional deletion of Brd4 in GCPs inhibits the sonic hedgehog pathway and SHH mediated proliferation. **a** Strategy utilized for deleting Brd4 conditionally in the cerebellum. Brd4$^{tm1a(EUCOMM)Wtsi}$ heterozygous mice were obtained through the Knockout Mouse Project Repository at Baylor University. These mice were bred to Tg (ACTFLPe) mice to create Brd4$^{fl/+}$ mice. Brd4$^{fl/+}$ mice were bred to Tg (Atoh1-cre) mice to obtain conditional cre expression in the granule cell lineage. Tg (Atoh1-cre)+/−;Brd4fl/+ mice were then backcrossed to generate Tg (Atoh1-cre+);Brd4$^{fl/fl}$ and Tg (Atoh1-cre-); Brd4$^{fl/fl}$ littermates. **b** GCPs from Tg (Atoh1-Cre+);Brd4$^{fl/fl}$ mice express less Brd4, Gli1 and Gli2 than Tg (Atoh1-Cre-);Brd4$^{fl/fl}$ mice. RNA was extracted from isolated GCPs from P8 mice, and qRT-PCR was performed and normalized to Gapdh. Primers flanking both flox sites were used to verify loss of Brd4 exon 5. **c** GCPs from Tg (Atoh1-Cre+);Brd4$^{fl/fl}$ mice have less total Brd4, phospho-Brd4 S492/494, and Cyclin D1 than Tg (Atoh1-Cre-);Brd4$^{fl/fl}$ mice. GCPs from P8 mice were isolated, and nuclear extracts or whole cell lysates were resolved by SDS-PAGE and western blot analysis. Gapdh was used as a loading control. **d** GCPs from Tg (Atoh1-Cre+);Brd4$^{fl/fl}$ mice proliferate less than Tg (Atoh1-Cre-);Brd4$^{fl/fl}$ GCPs. Following purification, GCPs were plated for 48 h with SHH, incubated with EdU and with or without SHH for 2 h, then plated with fresh media on poly-D-lysine/laminin coated coverslips for 3 h. Cells were fixed for EdU detection, reaggregates were imaged with a confocal laser-scanning microscope, and EdU positive cells were quantified with ImageJ. The results are the average of three independent experiments and are represented as the mean ± SEM. A one-way ANOVA followed by Tukey's multiple comparison testing (5B) or a two-way ANOVA followed by Bonferonni's multiple comparison testing (5D) was performed (*$p < 0.05$, **$p < 0.01$, ***$p < 0.001$, ****$p < 0.0001$, N.S. no significance). Source data can be found in source data graphs under tabs for Fig. 5b, d and in source data Fig. 5

sedimentation to yield an enriched GCP fraction[20]. The cells were then pre-plated on a Petri dish to remove contaminating glia. Purified GCPs were resuspended in medium (BME, 1.5% glucose, 20 mM glutamine, 10% horse serum, 5% fetal bovine serum, 1% penicillin/streptomycin) and then plated. For proliferation assays, GCPs were cultured in suspension in the presence of mouse recombinant SHH (0.25 ng/mL, 464-SH, R&D Systems); for cell-cycle exit and differentiation assay, GCPs were plated on poly-D-lysine (100 µg/ml, P6407, Sigma)-laminin (20 µg/ml, L2020, Sigma)–coated plates. For treatment with compounds, 100 nM of SR- 1277, the indicated amounts of I-BET151, or DMSO were added to the culture medium for 24 h.

**Cerebellar organotypic slice culture and treatment**. Cerebella were isolated from P8 CD1 mice. Sagittal slices (250 µm) of cerebellar cortex were generated using a Leica VT1000S vibratome, and slices were plated on Millipore culture inserts in six-well culture dishes containing 1.5 ml of serum-free medium (Basal Medium Eagle (Gibco), 0.45% D-(+)-glucose solution (Sigma), 1× ITS supplement (Sigma), 2 mM L-glutamine (Gibco), 100 U/ml penicillin/streptomycin (Gibco). The slices were then submerged in 2.5 ml of medium containing DMSO or I-BET151 (1 µM) for 24 h, after which 1 ml was removed so that the slices were no longer submerged, and the medium was below the insert.

**RNA sequencing**. Extracted RNA was sent to the John P. Hussman Institute for Human Genomics for sequencing. RNA quality was tested by ThermoScientific

NanoDrop or Agilent Bioanalyzer and confirmed to have RIN numbers > 8.5. Sequencing was performed on Illumina HiSeq2000 with three samples per lane, generating an average of 95 M 2 × 100 bp reads per sample. After quality filtering and trimming with FastQC remaining RNA-seq reads were aligned to the Ensembl mouse genome (v.87) using TopHat (v.2.1). For all samples, 80–90% of reads aligned successfully. Differential expression analysis was performed by CuffDiff 2.2 using the "classic-fpkm" parameter for the normalization method, and the "pooled" parameter for the dispersion method. Sequencing results from this study have been deposited in under accession number SRP146255.

**Functional annotation analysis**. For each STEM profile, we performed functional annotation analysis using DAVID[41] against the Level 5 Gene Ontology Biological Processes terms (GOTERM_BP_5). To adjust for the false discovery rate, we only considered terms with a Benjamini–Hochberg adjusted p-value of 0.05.

**Flow cytometry**. For flow-cytometric analysis, isolated GCPs were removed from dishes, washed with PBS and 1% BSA, and fixed with 10% ethanol in PBS overnight at 4 °C. GCPs were then stained with 69 µM propidium iodide in 38 µM sodium citrate buffer and 1 µM RNAse A at 37 °C for 30 min. The number of cells in G1, S, or G2 phases was determined using a fluorescence-activated cell-sorting device (LSRII, Becton Dickinson) and analyzed by FlowJo software.

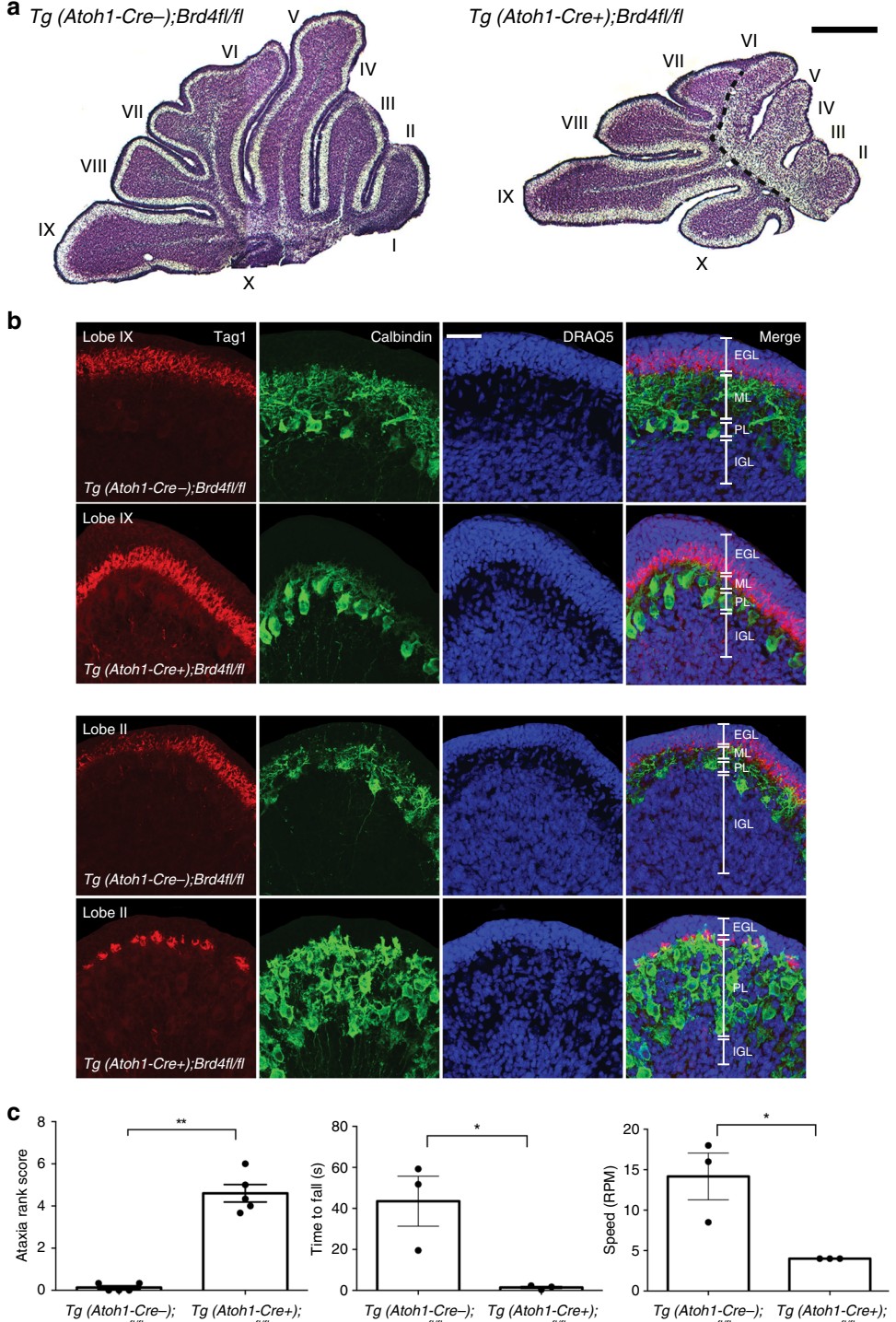

**Fig. 6** Conditional deletion of Brd4 in the developing cerebellum induces cerebellar deficits and ataxia. Tg (*Atoh1-Cre +*);*Brd4*$^{fl/fl}$ mice have disrupted cerebellar morphology relative to Tg (*Atoh1-Cre-*);*Brd4*$^{fl/fl}$ mice. Representative images are from P8 cerebellum mid-sagittal sections of approximately the same location in vermis. **a** Cresyl violet stain shows reduced cerebellum size in Tg (*Atoh1-Cre +*);*Brd4*$^{fl/fl}$ mice versus Tg (*Atoh1-Cre*);*Brd4*$^{fl/fl}$ mice. Cerebellar lobes are indicated I–X. Dotted line in Tg (*Atoh1-Cre +*);*Brd4*$^{fl/fl}$ divides the anterior and poster cerebellum to emphasize changes to proliferation and structure and proliferation, respectively, in each region. Scale = 500 μm. **b** Brd4 loss in GCPs disrupts Purkinje cell development. Confocal images are from the posterior lobe IX and anterior lobe II. Tag1 is a marker for migrating GCPs, Calbindin is a marker for Purkinje cells, and DRAQ5 is a DNA marker. Scale = 50 μM. EGL = external granule layer, ML = molecular layer, PL = Purkinje cell layer, IGL = internal granule layer. **c** Tg (*Atoh1-Cre +*);*Brd4*$^{fl/fl}$ mice exhibit symptoms of cerebellar ataxia. Adult (≥ P28) Tg (*Atoh1-Cre +*);*Brd4*$^{fl/fl}$ mice or Tg (*Atoh1-Cre +*);*Brd4*$^{fl/fl}$ mice were assessed for ataxia symptoms using multiple rank composite tests. Scores from each test were pooled with higher scores indicating symptom severity. Rotarod testing was performed, and latency to fall and speed at fall were recorded for each group after 5 days of training. The results are the average of three independent experiments, and are represented as the mean ± SEM. For ataxia, rank scores a Mann–Whitney U test and for rotarod testing an unpaired *t* test were performed (*$p <$ 0.05, **$p <$ 0.01). Source data can be found in source data graphs under tab for Fig. 6c

**[3]H-thymidine assay**. GCPs were plated ($3 \times 10^5$ cells/well) in 96-well dishes. DMSO or I-BET151 was added to the medium, and cells were maintained in culture for 24 h, in the presence of SHH. Then, 1 µCi [methyl-[3]H]-thymidine (Amersham) was added to each well, and cells were harvested 22 h later and analyzed using TopCount (Perkin–Elmer).

**EdU incorporation assay**. For the in vivo study, P8 CD1 mice were injected intraperitoneally twice with 10, 25, or 50 mg/Kg of JQ1 during 24 h. Two hours after the last injection, 50 mg/Kg of EdU was administered subcutaneously over the top of the neck. One hour after EdU administration, pups were perfused with 4% paraformaldehyde (PFA). Brains were processed for immunohistochemistry, and cerebella were cut in 20 µM sections with a cryostat. For the ex vivo proliferation assays, 1 ml of 25 µM EdU (Invitrogen) was added on top of the slices after 24 h in culture. Thus, the final concentration was 20 µM EdU per 2.5 -ml medium. Then, slices were fixed for 2 h with 4% PFA. For in vitro proliferation assays, GCPs were treated with EdU (20 µM) for 2 h. The cells were then washed with PBS, plated in poly-D-lysine/laminin-coated dishes for 2–3 h, and further fixed with 4% PFA/30% sucrose.

Then, sections, slices, or cells were permeabilized and stained using the Click-iT® EdU Alexa Fluor® 594 Imaging Kit (Invitrogen). Samples were imaged using the z-stack of a confocal laser-scanning microscope (Olympus, FV1000), and the images were analyzed using Fiji software (ImageJ).

**Immunohistochemistry**. Mice were perfused with 4% PFA, and cerebella were extracted. Cerebella were fixed in 4% PFA for 2 h, embedded in 30% sucrose in PBS, and cut into 20-µm sections with a cryostat (Leica). Sections were then permeabilized and blocked in 0.5% Triton X-100, 5% fetal bovine serum for 1 h at room temperature and incubated overnight at 4 °C with the following primary antibodies: rabbit anti-Brd4 (1/1000, Bethyl Laboratories Inc.), mouse or rabbit anti-calbindin (1/1000, Swant300, CB38), and mouse anti-Tag1 (1/1). The slices were then washed with PBS, and incubated for 4 h at room temperature with the following secondary antibodies as appropriate: Alexa Fluor® 488 goat anti–rabbit IgG, Alexa Fluor® 594 goat anti-mouse IgG or Alexa Fluor® 594 goat anti-mouse IgM (all 1/500, Invitrogen). Sections were then washed with PBS and incubated with Hoechst stain (Invitrogen) or DRAQ5 (AbCam) before mounting using ProLong Gold Antifade mounting medium (Invitrogen). Confocal images were acquired with a confocal laser-scanning microscope and were further analyzed with Fiji software (ImageJ).

**Protein extract preparation, antibodies, and western blot analysis**. Cells were homogenized, and extracts were prepared using lysis buffer (NER buffer from the NER-PER nuclear and cytoplasmic extraction reagents (Thermo Fisher Scientific), 1× protease inhibitor cocktail, 1 µM microcystin-LR). The soluble fraction was recovered by centrifugation at $16,000 \times g$ for 10 min at 4 °C. Protein concentration was measured with the BCA Protein Assay Kit (Pierce Biotechnology), and 30 µg of protein from each sample was resolved by SDS-PAGE. The resolved bands were transferred onto a nitrocellulose membrane and subjected to western blotting with the appropriate antibodies.

The following primary antibodies were used: mouse anti-CK1δ (C-8) (1/1000, Santa Cruz Biotechnology, sc-55553), goat anti-CK1δ (N-19) (1/500, sc-6475, Santa Cruz Biotechnology), rabbit-anti C-terminal Brd4 (1/2000)[19], rabbit anti-Brd4 (1/1000, A301-985A50, Bethyl Laboratories Inc.), rabbit anti-phospho-Brd4 492/494 (1/2000, ABE1453, Millipore), mouse anti-Gapdh (1/5000, NB300-221, Novus Biolechne), goat anti-CK2α (1/500, sc-6479, Santa Cruz Biotechnology), mouse anti-Flag (1/5000, A8592, Sigma), and rabbit anti-cyclin D1 (1/1000, ab134175, Abcam). The following secondary antibodies were used: anti-goat IgG–HRP (1/1000, 7074, Cell Signaling), anti-mouse IgG–HRP (1/1000, NXA931, GE Healthcare) and anti-rabbit IgG–HRP (1/1000, NA9340V, GE Healthcare).

**RNA isolation and qRT-PCR**. Cells were lysed in 1 mL of TriZol Reagent (Invitrogen), and the RNA was purified with the RNeasy Mini Kit (Qiagen). RNA was then reverse-transcribed with a High Capacity cDNA Reverse Transcipton Kit (Applied Biosystems). TaqMan probes were designed with the TaqMan Gene Expression Assay tool (Applied Biosystems). The qRT-PCR was performed using a TaqMan® Gene Expression Master Mix (Applied Biosystems) in a CFX384 TouchTM Real-Time PCR Detection System (Bio-Rad). Fold change in gene expression was estimated using the computed tomography comparative method and normalizing to the *Gapdh* computed tomography values and relative to control samples (Table 1).

**In vitro phosphorylation of Brd4 with CK1δ**. Purified bacterial Brd4 phosphorylation-dependent interaction domain (PDID) proteins were used for this study (PDID wild-type, PDID S484/488 A, PDID S492/494 A). In each reaction, 100 ng of PDID protein was incubated with 200 µM of ATP, 1000U of CSNK1D (NEB, P6030S), and 1X NEBbuffer for protein kinases for 30 min at 30 °C. Laemmli sample buffer was added to terminate the reactions, and the samples were heated to 95°C and resolved by SDS-PAGE. The resolved bands were transferred onto a nitrocellulose membrane and subjected to western blotting with the appropriate antibodies.

## Table 1 Accession numbers for genes used in quantitative PCR

| Gene | TaqMan® Gene Expression Master Mix (Applied Biosystems) accession number |
| --- | --- |
| *Cyclin B1* | Mm01322149_mH |
| *Atoh1* | Mm00476035_s1 |
| *Tuj1* | Mm00727586_s1 |
| *Gap43* | Mm00500404_m1 |
| *Gapdh* | Mm99999915_g1 |
| *Brd4, exon 4-5* | Mm01348074_m1 |
| *Brd4, exon 5-6* | Mm00480392_m1 |
| *Brd4, exon 7-8* | Mm00480394_m1 |
| *Brd2* | Mm01271171_g1 |
| *Brd3* | Mm00469733_m1 |
| *Gli1* | Mm00494654_m1 |
| *Gli2* | Mm01293117_m1 |
| *Gli3* | Mm00492337_m1 |
| *Cyclin A1* | Mm00432337_m1 |
| *Cyclin D1* | Mm00432359_m1 |

In the case of the radioactivity assay, 5 µCi [γ-[32]P]ATP (Perkin–Elmer, BLU002H250UC) was added to each reaction. The resolved SDS-PAGE gel was exposed, and radioactive signal was quantified using a Cyclone phosphor imaging system (Perkin–Elmer).

**Chromatin immunoprecipitation (ChIP)**. GCPs were collected, cross-linked with formaldehyde, and further lysed and sonicated. The chromatin was immunoprecipitated with antibodies for Brd4 (Bethyl Laboratories Inc., A301-985A50) and negative control antibody IgG (Abcam, ab37415). DNA–protein cross-links were reversed, and DNA was purified to be used in the quantitative amplification of *Gli1* locus with SYBR Green (−3250 ~−2501, forward primer, TGGCTCACAACCATC CTGTA, reverse primer, GAGATGCCCTTGCTTCTGTC; +1251 ~+2000, forward primer, ACCCAGGAATCCAAGGTGTC, reverse primer, TCCTGAAAGCAGGCAGTAGC) (Table 2).

**Generation of Brd4[fl/fl] mice**. Brd4[tm1a(EUCOMM)Wtsi] (MGI ID: 4441798) heterozygous mice were obtained through the Knockout Mouse Project Repository at Baylor University from Dr. John Seavitt. These mice were bred to B6.Cg-Tg (ACTFLPe)9205Dym/J mice (The Jackson Laboratory Stock #005703) to remove the neomycin resistance gene and LacZ reporter gene (forward primer, CTTGGGTGGAGAGGCTATTC, reverse primer, AGGTGAGATGACAGGAGA TC) to create Brd4[tm1c] heterozygous mice. In parallel, Brd4[tm1c] heterozygous mice were bred to homozygosity (hereafter referred to as Brd4[fl/fl]) or to B6.Cg-Tg (Atoh1-cre)1Bfri/J mice (The Jackson Laboratory Stock #011104, hereafter referred to as Tg (Atoh1-cre)) to obtain conditional cre expression in the granule cell lineage. Brd4[fl/fl] and Tg (Atoh1-cre +/−);Brd4[fl/+] mice were crossed to generate Tg (Atoh1-cre +/−);Brd4[fl/fl] or Tg (Atoh1-cre−/−);Brd4[fl/fl] littermates (Table 3). Tg (Atoh1-cre +/−);Brd4[fl/fl] do not breed successfully, therefore, Tg (Atoh1-cre) +/-;Brd4[fl/+] were maintained for experiments.

Genotype primers: Cre forward: AGAACCTGAAGATGTTCGCG; Cre reverse: GGCTATACGTAACAGGGTGT; Brd4 forward 1: TTTGACCTCTGCTCGTGTA GTG; Brd4 forward 2: ACCGCGTCGAGAAGTTCCTATT; Brd4 reverse: CATTG TACCCAGGGCTCCTTTCA.

**Behavioral testing**. For ataxia rank composite scoring, we followed the composite scoring method for ataxia outlined in Guyenet et al.[42] with modifications. Briefly, blinded researchers scored Tg (Atoh1-cre +/−);Brd4[fl/fl] or Tg (Atoh1-cre−/−); Brd4[fl/fl] mice on the ledge test, on the hindlimb clasping test, and on the gait test. These three composite tests are used most frequently to establish ataxia severity. Composite scores were averaged and compared in Tg (Atoh1-cre +/−);Brd4[fl/fl] or Tg (Atoh1-cre−/−);Brd4[fl/fl] mice. We used a rotarod paradigm to examine mouse balance, coordination and muscle function simultaneously. Mice were trained with rotarod twice daily for 5 days in the accelerating mode (5–40 rotations per minute over 2.5 min), and the latency to fall and speed were recorded after the training period. Average latency times and speed for the testing interval were compared in Tg (Atoh1-cre +/−);Brd4[fl/fl] or Tg (Atoh1-cre−/−);Brd4[fl/fl] mice.

**Differentially expressed gene analysis**. The gene-collapsed Affymetrix gene expression data were downloaded from GEO (GSE74400) and Differentially Expressed Genes (DEGs) compared with P0 were identified using limma[43]. The expression levels of subset of cell-cycle and neuronal-related genes belonging to the STEM-generated Profile 47 were then plotted (Supplementary Fig. 2) showing a

**Table 2 Primers for amplification of *Gli1* locus in the Chromatin immunoprecipitation assay**

| Locus amplified | Forward primer | Reverse primer |
|---|---|---|
| −3250 ∼ −2501 | TGGCTCACAACCATCCTGTA | GAGATGCCCTTGCTTCTGTC |
| +1251 ∼+ 2000 | ACCCAGGAATCCAAGGTGTC | TCCTGAAAGCAGGCAGTAGC |

**Table 3 Primers for genotyping transgenic mice for Cre and flox site**

| Type | Forward primer | Reverse primer |
|---|---|---|
| To remove the neomycin resistance gene and LacZ reporter | CTTGGGTGGAGAGGCTATTC | AGGTGAGATGACAGGAGATC |
| To create *Brd4tm1c* heterozygous mice | CTTGGGTGGAGAGGCTATTC | AGGTGAGATGACAGGAGATC |
| To genotype *Cre* mice | AGAACCTGAAGATGTTCGCG | GGCTATACGTAACAGGGTGT |
| To genotype *Brd4* mice | Forward 1: TTTGACCTCTGCTCGTGTAGTG | CATTGTACCCAGGCTCCTTTCA |
|  | Forward 2: ACCGCGTCGAGAAGTTCCTATT |  |

similar expression pattern to that from Supplementary Fig. 1. Furthermore, to quantify the overlap between the DEGs of the two data sets, we calculated the pairwise percentage of overlap among all time points, as shown in Supplementary Fig. 2. The percentage of overlap is defined as the number of DEGs that were common between the two data sets divided by the total number of DEGs in the corresponding RNA-seq Timeseries timepoint.

**Mouse embryonic fibroblast (MEF) cultures**. MEF *Sufu*$^{-/-}$ cells[11] were maintained in DMEM media with 10% neonatal calf serum and 1% penicillin/streptomycin. Overall, $1 \times 10^5$ cells per well were plated in a 12-well plate. In all, 100 nM of SR-653234 or SR-1277 CK1δ inhibitors were added into the media. DMSO was used as a vehicle control, 100 nM of GDC0449 as negative control, and 10 μM GANT-61 as positive control. Twenty-four hours later, RNA was extracted and SHH target gene expression was examined by Taqman probe-based qRT-PCR. Brd4 ChIP analysis was performed as described as above.

**siRNA transfections in GCP cultures**. GCPs were isolated from P6 CD1 pups, and $7.5 \times 10^5$ cells per well were plated in suspension in a 48-well plate. Cells were transfected with 1 μM SMARTpool Accell Mouse Brd4 siRNA (57261, E-041493-00, Dharmacon GE Helathcare) or 1 μM SMARTpool Accell Mouse GFP Scramble siRNA (D-001950-01-05, Dharmacon GE Helathcare) in low-serum media (Accell siRNA delivery media (Dharmacon GE Helathcare), 1.5% glucose, 20 mM glutamine, 2% horse serum, 1% fetal bovine serum, 1% penicillin/streptomycin) with mouse recombinant SHH (0.25 ng/mL). Cells were collected 72 h post transfection, and RNA isolation and qRT-PCR were performed as described.

**Statistical analysis**. All experiments were conducted independently at least three times. Statistical analysis was performed with Prism software (Graphpad). Figures 1b, c, 2c, e, 4a, b, f: one-way ANOVA followed by Bonferroni multiple comparison testing ($p < 0.05$). Figures 3b, e, 4d: paired $t$ test ($p < 0.05$). Figure 5b: one-way ANOVA followed by Tukey's multiple comparison testing ($p < 0.05$). Figure 5d: two-way ANOVA followed by Bonferonni's multiple comparison testing ($p < 0.05$). Figure 6c: for ataxia rank scores Mann–Whitney test ($p < 0.05$) and for rotarod testing unpaired $t$ test ($p < 0.05$).

**Reporting summary**. Further information on research design is available in the Nature Research Reporting Summary linked to this article.

## Data availability

The authors declare that all transcriptional data supporting the findings of this study are freely available from the databases provided within this paper. Sequencing results from this study have been deposited in under accession number SRP146255. All other data supporting the findings of this study are available from the corresponding author, Dr. Nagi G. Ayad, upon request.

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

## Acknowledgements

We acknowledge funding from the NIH via grants NS067289, R56102590 to N.G.A. We also acknowledge support from the Epigenetics Program of the Sylvester Comprehensive Cancer Center at the University of Miami. NIH grant CA103867, CPRIT grants RP190077 and RP180349, and Welch Foundation grant I-1805 to C-M.C. We thank Dr. David Rowich for providing the Atoh1-Cre mice. We thank the Moraes laboratory for providing assistance with the phosphorylation assays. We thank Dr. Bradner's laboratory for providing the JQ1 compound used in these studies. We thank all members of the Center for Therapeutic Innovation and the Lemmon-Bixby laboratories at the University of Miami for helpful discussions.

## Author contributions

C.P. designed research, performed experiments, developed methods, analyzed data, and wrote the paper. M.E.M. designed research, performed experiments, developed methods, analyzed data, and wrote the paper. V.S. designed research, performed experiments, developed methods, analyzed data, and wrote the paper. Y.F., J.L., J.M., J.R-B. and S.K.T. performed experiments. C.V. analyzed data. C.-M.C. provided reagents. J.K.L, D.L., D.J.R. and M.E.H. analyzed data and wrote the paper. J.C. designed research, analyzed data, and wrote the paper. N.G.A. designed research, analyzed data, and wrote the paper.

## Additional information

**Competing interests:** The authors declare no competing interests.

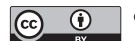

