## [Peer Review File · Nature Communications]

Reviewers' comments:

Reviewer #1 (Remarks to the Author):

In this paper the authors probe the role of Brd4 in regulating Shh-dependent proliferation of cerebellar granule cell precursors. Consistent with prior studies (Tang et al, Nature Medicine 2014; Long et al, JB, 2014) they show that Brd4 binds to the Gli1 promoter and regulates Shh-pathway genes. Although the previous studies were done in Shh-medulloblastomas, and the current studies are done in normal, developing GCPs, the additional insights from these results are limited. A novel feature of the manuscript is the demonstration that CDK1 delta phosphorylates Brd4, and that this is critical for Brd4 binding to the Gli1 locus. Surprisingly, although the authors use a conditional mutant in CDK1 delta for some of the biochemical studies, they do not demonstrate the in vivo phenotype of these mutants.

Specific issues.

1. In figure 1, the correlation between the downregulation of mitotic gene expression program and the upregulation of neuronal differentiation does not seem very tight, nor unexpected. It would be useful to probe these changes in greater depth.

2. In Figure 4, the authors show a less than expected change in development with conditional knock out of Brd4 in developing GCPs. One question from these results might be the time at which Brd4 protein is lost in the developing cells. The only time point shown is P8 (relatively late in the process). Is the limited (though clear) phenotype due to delayed loss of Brd4, or does this reflect a limited phenotype?

Reviewer #3 (Remarks to the Author):

Penas et al describe identifying a novel role for Brd4 in regulating proliferation of granule neuron progenitors in the developing cerebellum, through its casein kinase delta-mediated binding to the Gli1 locus. Experiments with new cerebellar progenitor-specific Brd4 knock out mice are convincing, as are slice culture experiments in Figure 3 using inhibitors. However, the initial premise is flawed by the lack of SHH in the Figure 1 experiments to examine cell cycle exit. SHH is an obligate mitogen for GCPs and it is well known that within hours of SHH derivation they will initiate reversible cell cycle exit. In vivo, GCPs exit the cell cycle despite ongoing exposure to SHH, therefore the experiments in Figure 1 should have been done in the presence of SHH to more closely recapitulate the in vivo situation.

The authors state that GCP cell cycle exit has not been well studied, but a simple search of the literature turns up reams of publications dating back decades examining this phenomenon, highlighting such players as p27, p18, arrestins, etc. Certainly role for Brd4 is novel, but this statement is misleading.

Overall organization: manuscript is missing page numbers, making it challenging to keep it organized and follow the flow. Figure are busy and overcrowded, and should be split up or moved to supplemental data.

Reviewer #4 (Remarks to the Author):

Transcriptome analysis of cerebellar granule cell progenitors prepared from 6-day old mice and exiting

the cell cycle to differentiate into neurons was performed. Time course analysis showed a downregulation of Gli1 expression and of the binding of the BET histone acetylation reader family member Brd4 to the Gli1 promoter within the first 2 hours of cell cycle exit. This was paralleled by a reduction of Brd4 phosphorylation. The role of CK1d in phosphorylating Brd4 at S492/494 was demonstrated using the specific inhibitor SR-1277, by Brd4 mutational analysis and by silencing CK1d. Treatment with the BET inhibitors I-BET151 or JQ1 reduced the proliferation of progenitor cells in vitro and in vivo. Conditional silencing of Brd4 in cerebellar granule cell progenitors in mice led to reduction of Gli1 and Gli2 expression, and diminished cell proliferation. This led to reduced cerebellar development and behavioral deficits.

The results are solid and interesting, and document in detail the role of Brd4 and Gli1/Gli2 regulation in neuronal differentiation. However several facets of this process have already been published (they are all referenced in the manuscript), admittedly in a different context and in less detail. For example, the role of BET proteins in neurogenesis and cell cycle progression of progenitor cells has been reported, including transcriptome analysis (Ref. 19). The involvement of Brd4 and the impact of BET inhibitors on medulloblastoma has been reported by several groups since 2013 (Ref. 4, 25, 26). Also, the role of Brd4 in Gli1/Gli2 expression is shown in these papers.

Reviewer #1 (Remarks to the Author):

In this paper the authors probe the role of Brd4 in regulating Shh-dependent proliferation of cerebellar granule cell precursors. Consistent with prior studies (Tang et al, Nature Medicine 2014; Long et al, JB, 2014) they show that Brd4 binds to the Gli1 promoter and regulates Shh-pathway genes. Although the previous studies were done in Shh-medulloblastomas, and the current studies are done in normal, developing GCPs, the additional insights from these results are limited. A novel feature of the manuscript is the demonstration that CDK1 delta phosphorylates Brd4, and that this is critical for Brd4 binding to the Gli1 locus. Surprisingly, although the authors use a conditional mutant in CDK1 delta for some of the biochemical studies, they do not demonstrate the *in vivo* phenotype of these mutants.

We thank the reviewer for their comments. However, we do not agree that the insights are limited. It is not obvious from the medulloblastoma studies that Brd4 would be required for normal cerebellar development. In fact, many medulloblastoma targets are not required for cerebellar development. Our results are striking as there are only a handful of genes that are required for normal cerebellar development. For instance, our prior studies deleted a critical cell cycle regulator involved in medulloblastoma growth, Cdh1 (Fzr1), with no effect on cerebellar development¹. Similarly, CK1 δ knockout, a kinase which has also an essential role in medulloblastoma, also had no effect on cerebellar development¹. We do not describe an *in vivo* phenotype for the Tg (*Atoh1-Cre*); *Cdk1d^{fl/fl}* mutant mice in this paper, because we have already published on this mouse line¹. While there does not appear to be a behavioral phenotype in Tg (*Atoh1-Cre+*); *Cdk1d^{fl/fl}* versus Tg (*Atoh1-Cre-*); *Cdk1d^{fl/fl}*, granule cell progenitors from Tg (*Atoh1-Cre+*); *Cdk1d^{fl/fl}* proliferate less in the presence of sonic hedgehog, which corroborates results from Tg (*Atoh1-Cre+*); *Brd4^{fl/fl}* mice presented in this manuscript. However, in an *in vivo* developmental setting, CK1 δ deletion is not lethal in the cerebellum. Therefore, it is essential to study genes *in vivo* to gain an understanding of their role in development. The Tg (*Atoh1-Cre+*); *Brd4^{fl/fl}* mice presented cerebellar ataxia, thus linking Brd4 deletion with a developmental defect, which has not been previously described.

Specific issues.

1. In figure 1, the correlation between the downregulation of mitotic gene expression program and the upregulation of neuronal differentiation does not seem very tight, nor unexpected. It would be useful to probe these changes in greater depth.

We thank the reviewer for their comments. We show a more detailed description of the gene expression changes that occur during cell cycle exit and neuronal differentiation in the current version of the paper. Our short time series expression modeling enabled us to parse changes to gene expression and each cluster with cellular processes such as mitosis (Figure 1, cluster 47) and neuronal differentiation (Figure 1, cluster 75). A critical aspect of our analysis is that *Gli1*, which is part of cluster 47 that is linked to granule cell progenitor proliferation, decreases very early in the cell cycle exit process. This occurs within two hours. The kinetics of *Gli1* decrease is related to loss of both Brd4 phosphorylation at serine 492/494 and the CK1 δ kinase. Using this modeling, in conjunction with our work on CK1 δ degradation and cell cycle exit, Brd4 regulation of *Gli1* expression, and other studies on Brd4 phosphorylation and chromatin binding, we hypothesized that there was a relationship between genes in cluster 47 with decreased CK1 δ levels and Brd4 phosphorylation. Indeed, we show in Supplemental Figure 1 that the consensus RNA signature cluster 47, coincides with Brd4 serine 492/494 phosphorylation loss. We would like to point out that this is first time Brd4 phosphorylation and binding to chromatin has been analyzed within the context of cell cycle exit. Our results demonstrate that Brd4 phosphorylation loss due to decreased CK1 δ levels is an early event in cell cycle exit of GCPs.

To further study gene expression during these processes, we have now compared cell cycle exit kinetics *in vitro* and *in vivo* by comparing our RNA-sequencing data with Zhu et al., (Neuron, 2016)², which analyzed the transcriptome of granule cells *in vivo*. We now show in a new Supplemental figure 2 that the *in vitro* cell cycle exit kinetics we observe *in vitro* are recapitulated *in vivo* after postnatal day 7.

2. In Figure 4, the authors show a less than expected change in development with conditional knock out of Brd4 in developing GCPs. One question from these results might be the time at which Brd4 protein is lost in the developing cells. The only time point shown is P8 (relatively late in the process). Is

the limited (though clear) phenotype due to delayed loss of Brd4, or does this reflect a limited phenotype?

We thank the reviewer for their comments. We have included additional immunohistochemistry data of Tg (*Atoh1-Cre+*); *Brd4*^{fl/fl} and Tg (*Atoh1-Cre-*); *Brd4*^{fl/fl} before (postnatal day 0 and postnatal day 3) and after (postnatal day 9) the postnatal day 8 time point shown in figure 6 (original figure 4). We have included sample sagittal images from approximately the same anatomical region as in postnatal day 8, mid-vermis, well as anterior and posterior cerebellar lobes, in Supplemental figures 7 and 8, respectively. From these data, the cerebellar morphology at postnatal day 0 in both the anterior and posterior regions remains intact (compare Tg (*Atoh1-Cre+*); *Brd4*^{fl/fl} and Tg (*Atoh1-Cre-*); *Brd4*^{fl/fl}). By postnatal day 3 cerebellum morphology diverges between Tg (*Atoh1-Cre+*); *Brd4*^{fl/fl} and Tg (*Atoh1-Cre-*); *Brd4*^{fl/fl} mice, where Tg (*Atoh1-Cre+*); *Brd4*^{fl/fl} cerebella maintain an underdeveloped anatomy, which is more severe in the anterior lobes. This underdevelopment, particularly in cell layer formation and dendritic outgrowth persists throughout the developmental points shown, and most likely account for the aberrant behavioral phenotype observed. Therefore, we conclude that this phenotype is not limited, but rather is a persistent developmental defect at and after postnatal day 3.

Reviewer #3 (Remarks to the Author):

Penas et al describe identifying a novel role for Brd4 in regulating proliferation of granule neuron progenitors in the developing cerebellum, through its casein kinase delta-mediated binding to the Gli1 locus. Experiments with new cerebellar progenitor-specific Brd4 knock out mice are convincing, as are slice culture experiments in Figure 3 using inhibitors. However, the initial premise is flawed by the lack of SHH in the Figure 1 experiments to examine cell cycle exit. SHH is an obligate mitogen for GCPs and it is well known that within hours of SHH derivation they will initiate reversible cell cycle exit. *In vivo*, GCPs exit the cell cycle despite ongoing exposure to SHH, therefore the experiments in Figure 1 should have been done in the presence of SHH to more closely recapitulate the *in vivo* situation.

We thank the reviewer for their comments. The purpose of our experiments is to examine the differential changes in gene expression that occur with granule cell progenitor cell cycle exit and eventual differentiation into mature granule cells. By examining differential gene expression during this process, we were able to determine not only that cell cycle gene downregulation occurs with concomitant neuronal gene upregulation, but also that Brd4 phosphorylation at serines 492/494 plays an important role in maintaining granule cell progenitor proliferation. Because sonic hedgehog is an obligate mitogen for granule cell progenitors, if we exposed granule cell progenitors just to sonic hedgehog, then as the reviewer points out, there would be proliferation. However, proliferation would be asynchronous, so we would not be able to obtain an accurate picture of cell cycle exit gene expression kinetics. In addition when plated on poly-d-lysine/laminin, granule cell progenitors do not proliferate indefinitely in the presence of sonic hedgehog and will eventually exit the cell cycle. When they are plated on poly-d-lysine/laminin coated plates with no signaling factors, we show in Figure 1 that granule cell progenitors undergo cell cycle exit and differentiation synchronously over a defined time period, thereby allowing us to perform the short time-series modeling analysis that is critical for our findings. We would like to point out that our *ex vivo* organotypic slice system has all factors including Shh present and therefore the defects in proliferation when we inhibit Brd4 are occurring in a system with endogenous Shh signaling present.

The authors state that GCP cell cycle exit has not been well studied, but a simple search of the literature turns up reams of publications dating back decades examining this phenomenon, highlighting such players as p27, p18, arrestins, etc. Certainly a role for Brd4 is novel, but this statement is misleading.

We thank the reviewer for their comments. While we do not dispute the players involved in cell cycle exit highlighted above, there has not been a detailed analysis of cell cycle exit in GCPs using RNA-sequencing. The closest is a recent study by the Hatten laboratory where the expression of genes during postnatal development was described². We compared our time-series model to the data presented in the Zhu et al., 2016 paper and found that the cell cycle genes in Cluster 47 in our *in vivo* studies decrease after postnatal day 7 *in vivo*. Therefore, we now have a means of comparing *in vitro* time series of GCPs with *in vivo*

cell cycle exit and differentiation. Another important aspect of our work is that we are the first to report that Brd4 phosphorylation is correlated with cell cycle proliferation in GCPs and provide a mechanism of how this is downregulated during cell cycle exit via loss of CK1 δ . This study prompted us to study the role of Brd4 during development, and we found that Brd4 is essential for a proper cerebellar development. Brd4 knockout leads to cerebellar ataxia. Thus, our studies are novel, and will be useful to define an appropriate time window for the treatment of children with medulloblastoma with BET inhibitors, to minimize functional deficits that may appear with pharmacological treatment.

Overall organization: manuscript is missing page numbers, making it challenging to keep it organized and follow the flow. Figure are busy and overcrowded, and should be split up or moved to supplemental data.

We thank the reviewer for their comments. We have reorganized our main figures so that the data are now divided among six, rather than four figures. We have added page numbers to make the manuscript more reader friendly.

Reviewer #4 (Remarks to the Author):

Transcriptome analysis of cerebellar granule cell progenitors prepared from 6-day old mice and exiting the cell cycle to differentiate into neurons was performed. Time course analysis showed a downregulation of Gli1 expression and of the binding of the BET histone acetylation reader family member Brd4 to the Gli1 promoter within the first 2 hours of cell cycle exit. This was paralleled by a reduction of Brd4 phosphorylation. The role of CK1d in phosphorylating Brd4 at S492/494 was demonstrated using the specific inhibitor SR-1277, by Brd4 mutational analysis and by silencing CK1d. Treatment with the BET inhibitors I-BET151 or JQ1 reduced the proliferation of progenitor cells *in vitro* and *in vivo*. Conditional silencing of Brd4 in cerebellar granule cell progenitors in mice led to reduction of Gli1 and Gli2 expression, and diminished cell proliferation. This led to reduced cerebellar development and behavioral deficits.

The results are solid and interesting, and document in detail the role of Brd4 and Gli1/Gli2 regulation in neuronal differentiation. However several facets of this process have already been published (they are all referenced in the manuscript), admittedly in a different context and in less detail. For example, the role of BET proteins in neurogenesis and cell cycle progression of progenitor cells has been reported, including transcriptome analysis (Ref. 19). The involvement of Brd4 and the impact of BET inhibitors on medulloblastoma has been reported by several groups since 2013 (Ref. 4, 25, 26). Also, the role of Brd4 in Gli1/Gli2 expression is shown in these papers.

We thank the reviewer for their comments. Our study is novel because we are examining the role of Brd4 during normal cerebellar development. Li et al. (old Ref. 19) show that BET protein inhibition plays a role in promoting neuron fate from undefined neural progenitors and in suppressing cell cycle progression *in vitro*. However, our findings are not based only on *in vitro* studies, but also on *in vivo* studies. There are very few genes that are essential *in vivo* in the developing cerebellum. APC/C-Cdh1 and CK1 δ inhibition *in vitro* reduces cell proliferation but deletion of Cdh1 or CK1 δ *in vivo* in the cerebellum yields no phenotype. Therefore, the observation that Brd4 is a novel regulator of cerebellar development is important. We also found that the Brd4 knockout leads to cerebellar ataxia, which has not been previously described before. We also describe that Brd4 activity is modulated by CK1 δ phosphorylation during cell cycle exit *in vitro* and *in vivo*, which has not also been described before.

While both Brd4 and BET inhibition have been shown to affect sonic hedgehog signaling in medulloblastoma, there has been no previous examination of the contribution of Brd4 to normal granule cell progenitor proliferation and cell cycle exit nor of its regulation of sonic hedgehog signaling during normal granule cell proliferation and cell cycle exit.

We have now analyzed the Brd4 knockout phenotype further and can demonstrate that the cerebellar defect in the Brd4 knockout occurs after P0 at P3. This is when Sonic signaling increases during GCP development. Therefore, we have defined a critical window when Brd4 is required in the developing nervous system, which has not been previously reported.

We also provide new data demonstrating that Brd4 knockout leads to loss of several cell cycle genes including Brd2. Brd2 is thought to compensate for Brd4 loss in some contexts but in the developing cerebellum Brd4 loss is not compensated by Brd2.

Finally, we have also now compared cell cycle exit kinetics *in vitro* and *in vivo* by comparing our RNA-sequencing data with Zhu et al., (Neuron, 2016)², which analyzed the transcriptome of granule cells *in vivo*. We now show in a new Supplemental figure 2 that the *in vitro* cell cycle exit kinetics we observe *in vitro* is recapitulated *in vivo* after postnatal day 7. Therefore, our studies contribute to an understanding GCP of cell cycle exit during the developing nervous system, which was not previously appreciated.

References

- 1 Penas, C. *et al.* Casein kinase 1delta is an APC/C(Cdh1) substrate that regulates cerebellar granule cell neurogenesis. *Cell Rep* **11**, 249-260, doi:10.1016/j.celrep.2015.03.016 (2015).
- 2 Zhu, X. *et al.* Role of Tet1/3 Genes and Chromatin Remodeling Genes in Cerebellar Circuit Formation. *Neuron* **89**, 100-112, doi:10.1016/j.neuron.2015.11.030 (2016).

REVIEWERS' COMMENTS:

Reviewer #3 (Remarks to the Author):

The authors have added important data to their manuscript including details on gene expression changes, IHC data and more complete analysis of the Brd4 ko phenotype. Altogether this makes a solid story about the role and regulation of Brd4 in developing GCPs.

Minor points:

Line 90: fast-track designation (and not approval)

Line 93, please add: in mice

Response to reviewer comments:

We thank the reviewer for their positive assessment of our manuscript. Our response to the reviewer's comments are below.

Reviewer #3 (Remarks to the Author):

The authors have added important data to their manuscript including details on gene expression changes, IHC data and more complete analysis of the Brd4 ko phenotype.

Altogether this makes a solid story about the role and regulation of Brd4 in developing GCPs.

Minor points:

Line 90: fast-track designation (and not approval)

We modified this to designation.

Line 93, please add: in mice.

We added this to Line 93.